# Translational Bioinformatics for Human Reproductive Biology Research: Examples, Opportunities and Challenges for a Future Reproductive Medicine

**DOI:** 10.3390/ijms24010004

**Published:** 2022-12-20

**Authors:** Kun Liu, Yingbo Zhang, César Martin, Xiaoling Ma, Bairong Shen

**Affiliations:** 1Reproductive Medicine Centre, The First Hospital of Lanzhou University, Lanzhou 730000, China; 2Biofisika Institute (UPV/EHU, CSIC), 48940 Leioa, Spain; 3Department of Biochemistry and Molecular Biology, University of the Basque Country UPV/EHU, 48080 Bilbao, Spain; 4Institutes for Systems Genetics, West China Hospital Sichuan University, Chengdu 610041, China

**Keywords:** reproductive medicine, translational bioinformatics, infertility, biomarkers, artificial intelligence, database, knowledge graph

## Abstract

Since 1978, with the first IVF (in vitro fertilization) baby birth in Manchester (England), more than eight million IVF babies have been born throughout the world, and many new techniques and discoveries have emerged in reproductive medicine. To summarize the modern technology and progress in reproductive medicine, all scientific papers related to reproductive medicine, especially papers related to reproductive translational medicine, were fully searched, manually curated and reviewed. Results indicated whether male reproductive medicine or female reproductive medicine all have made significant progress, and their markers have experienced the progress from karyotype analysis to single-cell omics. However, due to the lack of comprehensive databases, especially databases collecting risk exposures, disease markers and models, prevention drugs and effective treatment methods, the application of the latest precision medicine technologies and methods in reproductive medicine is limited.

## 1. Introduction

Reproduction is one of the most common phenomena in the development of the species, but the rise of reproductive medicine came with the birth of the world’s first test tube baby in 1978. Over the past 40 years, many new techniques and discoveries have emerged in reproductive medicine, not only in terms of clinical diagnosis and treatment, but also in basic theoretical research and the application of reproduction. Reproductive medicine is inevitably a major theme in the development of medicine as it relates to germ cells, which are central to the transmission of genetic information from one generation to the next. Thereby, the leapfrogging process of genetics was also a time of rapid growth in reproductive medicine.

With the completion of the Human Genome Project, we have moved from the genomic era to the post-genomic era. If the genomic era is the journey from traditional biology to bioinformatics, then the post-genomic era is the journey from bioinformatics to translational bioinformatics (TBI), that is, the development of methods to store, analyze and interpret relevant methods to transform vast amounts of biomedical and genomic data into predictive, preventive and proactive health engagement applications [1]. Therefore, understanding the development and current status of reproductive bioinformatics and integrating traditional biology, different levels and independent bioinformatics can help to realize a complete set of top-down or bottom-up regulatory networks [2], tap into the deep phenotypes of reproductive and individual health risks, and form a structurally complete system of basic and clinical inter-translation of molecular biology information network disease, which is of great importance for the precise management of human health. Here, we provide an overview of the current status of reproductive bioinformatics and highlight the challenges and opportunities for the development of translational reproductive bioinformatics.

## 2. Identification and Molecular Regulation of Biomarkers of Infertility and Reproductive Disease

With the development of high-throughput technologies, attempts have been made to introduce molecular diagnostic techniques into medical practice [3]. Genomics and functional genomics provide thousands of molecular markers of human reproduction and infertility, making possible molecularly oriented diagnosis and treatment as well as disease prediction and prevention in clinical medicine, facilitating more precise and personalized clinical decision making. Traditionally, researchers have used genetic or biomolecular markers to distinguish specific biological phenotypes [4]. This class of markers has greatly improved disease diagnosis, but can lack precision, early discovery, prognosis and treatment prediction. As the development of omics has generated massive amounts of multidimensional data, it is necessary to use mathematical, computer science and physical methods to integrate important information from multidimensional data into comprehensive map of the way components [5], and to identify regulatory networks and potential key players through continuous machine learning (ML) from this static network. These marker molecules and their expressions are then used to identify the corresponding phenotype (e.g., normal or disease) in a predictive step [4], or to identify drug target interactions and predict therapeutic responses. In general, the manifestation of diseases, including reproductive disorders, is not from the malfunction of a certain molecule, but from failure of multiple molecules and correlated systems or networks. Thus, the integration of omics, systems biology and bioinformatics with biology facilitates the elucidation of the synergistic relationships among multiple factors and complex networks of molecular interactions, to reveal the behavior of the system as a whole and to unravel complex characterization of diseases.

### 2.1. Development of Male Reproductive and Infertility Biomarkers

Growing clinical and epidemiological studies suggest that a synchronized increase in the incidence of male reproductive problems, such as genital abnormalities, testicular cancer, reduced semen quality and subfertility [6,7]. However, unexplained infertility in nearly 75% of cases [8] with normal routine semen analysis indicates that the concentration, viability and morphology of sperm currently practiced clinically do not accurately reflect sperm function [9]. Therefore, it is necessary to develop effective biomarkers that accurately reflect male reproductive health status and fertility. At present, prostate cancer (PCA) is the mostly studied area in male reproductive biomarkers, followed by abnormal sperm and seminal plasma function.

### 2.2. Biomarks for Prostate Cancer

PCA is the second most common cancer among males worldwide, and even has the highest morbidity and mortality rates among some European and African races [10,11]. Genes associated with copy number variation in PCA cancer were more likely to be found on chromosome (chr)8, chr17 and chr10, while genes associated with methylation variation were more likely to be found on chr1, chr19 and chr17 [12]. Meanwhile, X-linkedness of PCA was observed in North America, Finland, Sweden and Germany [13,14]. In general, autosomal dominant cases are more likely to be seen in younger adults and present familial clustering, whereas autosomal recessive cases and X-linked cases are sporadic with an older age of onset [15,16]. To date, there have been numerous studies of biomarkers for primary and metastatic PCA based on single- and multi-omics, with both shared salient genetic characteristics [17] and differences across the ethnic groups [18,19]. Moreover, genetic heterogeneity was also seen at multiple levels including age [20], Gleason pathology classification [21] and prognostic risk [22,23]. The identification of molecular subtypes of PCA based on The Cancer Genome Atlas (TCGA) [24,25] can be used as drug targets for precise, biomarker-driven treatment options. Furthermore, PCA causes germline genetic mutations. Conversely, rare germline variants are associated with rapid biochemical recurrence [26] and poor prognosis [27] after radical prostatectomy. The homeobox transcription factor gene homeobox B13 (HOXB13) is the only gene currently found to only be associated with hereditary PCA. BRCA mutation has been a major cause of hereditary breast and ovarian cancer. However, studies showed that male-carriers of BRCA1/2 mutations had an increased risk of PCA with a more aggressive phenotype [28]. Additionally, family history of breast cancer was found to be associated with elevated risk of PCA [29]. As a result, HOXB13, BCRA1/2 are the three basic genes recommended in PCA screening program guidelines [30]. ATM is another combined screening gene usually for prognostic evaluation and targeted therapy [31,32]. In addition, CHEK2, EPCAM, MLH1, MSH2, MSH6, PALB2, NBN, PMS2, RAD51D and TP53 can be chosen depending on the age, whether they have metastatic and family history, etc. [33].

Prostate-specific antigen (PSA) was the first biomarker to be used clinically for PCA screening, but its under- and over-diagnosis poses drastic limitations for clinical application. Therefore, current guidelines do not recommend PSA screening for the general population, but only for BRCA2 carriers [33]. Combined PSA and multiparametric magnetic resonance imaging (mpMRI) screening for BCRA carriers over 55 years of age can improve the diagnostic accuracy [28]. In addition, more than a dozen biomarkers such as lnRNA, mRNA, protein, amino acid and cfDNA derived from urine, blood, tissue and biopsy samples have now been commercially developed, reflecting more advantages not only in improving detection accuracy, but also in assessing the prognostic value of tumors in aggressive, metastatic or non-metastatic tumors, and in monitoring treatment outcomes [34]. However, due to cost and local availability constraints, it is currently difficult to be commonly performed in the clinic [34].

Advances in clinical and experimental techniques have led to a deeper understanding of PCA occurrence and progression, and have guided the development of advanced and accurate early detection patterns and targeted therapies. Computer algorithms and neural networks have facilitated the use of artificial intelligence (AI) in PCA diagnosis [35], Gleason grading [36], identification of biopsy pathology sections [37], augmented reality microscopy [38] and patient management [39], facilitating clinical decision making and improving personalized management of PCA.

In the last decade, new therapies have been approved for early localized [40] and advanced PCA [41], as well as evolving therapies such as cancer vaccines, chimeric antigen receptor T (CAR-T) cells, bispecific T cell engagers (BiTEs), other targeted agents (e.g., AKT inhibitors) and various combination therapies [41] have marked an era of precision medicine for PCA. However, there are still some challenges in the application of these biomarkers and therapies in preclinical or clinical practice, such as the need for more high-quality evidence for existing therapies due to the heterogeneity of tumors, the need for systemic therapies that take into account systemic factors [42], the timing and potential cooperation of multiple driver markers, and multiple drug resistance mechanisms, among others [43].

### 2.3. Biomarkers for Male Infertility

The rapid decline of the world birthrate is a serious societal problem. In addition to economic and behavioral factors, biological factors are another crucial impactor [44,45]. That is, human fertility is declining and will continue to fall [46]. As far as human infertility is concerned, the male factor cannot be ignored (Figure 1). Studies have shown that men with impaired reproductive health (including poorer semen parameters and lower testosterone levels) have decreased general health [47], increased risk of tumors [48,49] and mortality [50,51], as well as increased female pregnancy miscarriages and birth defects in offspring [52,53]. Therefore, early detection of male infertility is important not only for early identification and correction of male fertility and fecundity, but also for the opportunity to improve the medical status of male general health and well-being [54], and even to improve the physical and mental health of women and future generations.

Chromosomal abnormalities, Y chromosome microdeletions and single gene mutations are the three major genetic factors responsible for male infertility. Of these, azoospermia is most significantly associated with genetic abnormalities (25%) [55]. In 1976, Tiepolo and Zuffardi first discovered that Yq11 was associated with spermatogenesis [56]. In the 1990s, further analysis of the Yq11 region identified proximal, intermediate and distal segments, defined respectively as azoospermia factor (AZF)a,b,c [57,58]. Among them, AZFa and AZFb deletions are often infertile due to azoospermia, while AZFc deletion can manifest only as oligospermia [59,60], and reproductive risk is passed on to offspring due to microinjection of a single sperm during assisted reproduction technology (ART). Mutations in the cystic fibrosis transmembrane conductance regulator (CFTR) are the pathogenic basis of cystic fibrosis (CF) [61] and can lead to obstructive azoospermia (OA) due to 60–70% congenital bilateral absence of the vas deferens (CBAVD), which accounts for 1–2% male infertility and is the only known genetic cause of CBAVD [62], and is the only genetic factor at play in CBAVD [63]. The study identified 17,754 genes associated with male reproductive defects, involving 13 different phenotypes such as spermatogenesis, male reproductive system development and sex hormone regulation. In addition to the sex chromosomes, all other 22 pairs of autosomes are distributed with genes associated with male infertility, with chr X, Y, 6, 17, 19 and 20 all containing more than 10% of the genes associated with infertility [64]. However, only 1564 genes have been confirmed, and only 120 of these genes have been assessed by the International Male Infertility Genomics Consortium (IMIGC) for moderate, strong or clear associations with 104 infertility phenotypes [65,66]. This is partly due to the fact that the underlying genetic causes of male infertility are the result of the accumulation of many rare genetic events, especially a large number of common variants with small effect sizes that have been screened for since the application of sequencing technology [67]. On the other hand, the cost of investigation and the large amount of data required by bioinformatic analysis have led to the problem of “small n (number of patients) and large p (parameters)” in big data [68], leading to unstable conclusions. Understanding the genetic factors of male reproduction and fertility helps clinical etiology diagnosis rather than diagnosis for pathophysiological phenomena. Professional genetic counseling helps clinical decision making by reasonably predicting the likelihood of sperm extraction from the testes during reproductive age, using preimplantation genetic testing (PGT) to block genetic defects, and developing pharmacogenetics to provide safer, more effective and economical drug treatment options for patients.

Compared with genomics, other omics technologies such as transcriptome, epigenome, proteome and metabolome are more complicated. One is that these data are high-dimensional and functionally complex. Secondly, they are dynamically changing and there is a large variability in their expression under different conditions. Thirdly, different experimental methods and bioinformatics computational methods may lead to inconsistent experimental results [69]. Therefore, there is an urgent need to integrate bioinformatics data on the interactions between gene expression, mutations and pathways to identify effective biomarkers for the early detection and prognosis of reproduction and fertility.

The RNA content of spermatozoa is extremely minimal, about 10–20 fg/sperm, and the total amount is only 1% of that of somatic cells [70]. However, there is a wide variety of species, a large amount of transcripts and complex biological functions. The utility of sperm RNA as a biomarker of infertility has been explored and is involved in several biological processes such as spermatogenesis, sperm movement, morphogenesis, capacitation, fertilization, early embryogenesis and transgenerational epigenetic transmission [71,72,73]. In 2002, Ostermeier [74] used a computerized Boolean search strategy to match testis microarray results with the UniGene database and identified 2416 new testis-expressed genes. Subsequently, further data mining of the testes, pooled ejaculate and single ejaculate spermatozoa of fertile males identified 7157, 3281 and 2780 unique expressed sequence tags (ESTs), respectively. The automated program Onto Express was used to systematically convert genetic fingerprints into functional atlas to determine the gene ontology and corresponding gene function for each EST mapping, and mRNA was found to be consistent in mature sperm with that in testis, suggesting that sperm RNA can be used as a noninvasive surrogate for testis-specific infertility and toxicology investigations [75]. Lee and Luk et al. [76,77] successively developed the web-based GermSAGE and GermlncRNA databases based on public expression databases. In recent years, the development of single-cell [78] and spatial transcriptomics [79] has further revealed stage-dependent gene expression analysis of testicular cells. The four cellular states of human spermatogonial stem cells (HSSC) [78], from quiescent HSSC to proliferating, metabolically active, differentiated spermatogonia, and the subsequent sequential and progressive development of spermatogonia and spermatocytes during spermatogenesis, are biologically important, and the spatial interactions of different subpopulations of these molecules are dissected. These studies provide valuable tools to explore some of the potential pathways of male infertility, to better understand the genetic networks that regulate pathophysiological changes in sperm, to identify new genes and new biomarkers, and to provide clues to predict residual spermatogenic sites in non-OA patients during ART [80].

As is well known, proteins are the direct performers of the functions of life activities, and metabolites are functionally regulated active substances that modulate protein interactions, alter enzyme activity, change protein stability and subsequently regulate body metabolism. Infertility is often caused by complex interactions between genetic susceptibility, environmental factors and lifestyle choices. Therefore, the application of functional genomics technologies such as proteomics and metabolomics has revealed many underlying molecular changes directly involved in the relevant pathophysiological processes, establishing links between genotype and phenotype and facilitating the translation of esoteric and complex digital codes into clinical practice. Proteomics has now uncovered protein expression throughout the sperm and its subcellular structures and, is being used to identify specific protein markers for various reproductive diseases [81,82,83]. Seminal plasma, a mixture of secretions from several male accessory glands, is not only a carrier of spermatozoa but also provides them with nutrients and plays an important role in semen coagulation and liquefaction as well as in sperm motility and fertilization. Seminal plasma also regulates sperm function and its physiological activity in the female reproductive tract, especially the interaction of the female reproductive tract immune system. Therefore, the study of seminal plasma protein and metabolic profiles can help to clarify its correlation with semen parameters (such as abnormal semen liquefaction, sperm concentration, sperm count, motility and morphology) [84] and to select biomarkers of male fertility disorders. Moreover, data from seminal plasma to understand the presence of proteins and metabolites from the testes, epididymis, seminal vesicles and prostate, enable the non-invasive detection of prostatitis, varicocele, and diseases such as prostate cancer and benign prostatic hyperplasia. It is also an important avenue for virology and drug toxicology research [85,86].

### 2.4. Development of Female Reproductive and Infertility Biomarkers

Infertility is a highly clinically heterogeneous condition. Apart from the obvious tubal, ovulatory and male sperm factors, many patients do not acknowledge or even realize infertility and instead are persistent in the correction of the primary disorder. However, so far, there are no effective etiology treatments specific for the infertility-related chronic diseases such as polycystic ovarian syndrome (PCOS), endometriosis, premature ovarian insufficiency (POI) and amenorrhea (Figure 2); only symptomatic treatment can be provided. Secondly, 15% of cases remain unexplained [87]. Thus, mining biomarkers and pathways of infertility and its related chronic diseases can provide comprehensive understanding of the deep phenotypes of infertility, and the common and differential characters among them, providing personalized treatment and effective prediction.

### 2.5. Hereditary Factors

Many of the chronic diseases associated with female infertility have a strong genetic component. Genome-wide sequencing can identify common genetic variants. In 2010, the first whole genome sequencing (WGS) was performed on endometriosis to explore its genetic complexity [88]. Nineteen genetic variants on chr 1, 2, 6, 7, 9 and 12 have been reported in several studies and are significantly associated with clinical staging (stage III/IV, rather than stage I/II, stage B versus stage A disease were 31 versus 15%) [89,90,91]. In 2011, the WGSs of Chinese Han [92] and Caucasian [93] women with PCOS were successively published, showing that chr 2p16.3 (rs13405728), 2p21 (rs13429458) and 9q33.3 (rs2479106) were strongly associated with PCOS. Even if the diagnostic criteria are different, PCOS still has the same genetic background [94]. The 24 candidate loci [95,96] comprised a hierarchical signaling network, adapter proteins and converging associated downstream signaling cascades [97], leading to multiple phenotypes including abnormal LH and T secretion, high AMH, insulin resistance and follicular dysplasia [96,98]. In 2013, Elashoff et al. used WGS for the first time to identify nearly 50,000 de novo variants in women with infertility and POI [99]. The incidence of exonic copy number variation (CNV) associated with female infertility is about 4%, of which about 20% are located on chr X and 11% on chr 16 [100]. It follows that reproductive disorders associated with female infertility are often not the result of variations in a single candidate gene, but rather the interactions of multiple genes with each other and with the environment. Similarly, the same candidate loci may share the similar regulatory mechanisms in male infertility [101,102], and may also cause oocyte maturation disorder, abnormal zona pellucida and aberrant meiosis, manifesting as fertility disorders such as embryonic arrest, recurrent miscarriages or repeated embryo implantation disorders during ART [103,104,105]. There may also be a common genetic basis for clinical phenotypes [106,107,108] such as obesity, menstrual disorders, pelvic floor dysfunction, lower fetal birth weight and other systemic or psychological disorders [109,110,111,112,113] such as tumors, cardiovascular disease, diabetes and depression.

In addition, genome-wide sequencing is used not only to diagnose known reproductive disorders, but also to assess the risk of chromosomal abnormalities and single gene disorders that are clearly identified at the preconception stage (i.e., PGT) [114,115,116] and also to facilitate the identification of multifactorial genetic predispositions of sub-fatal conditions [117,118], including those affecting reproductive health. The acquisition and collation of reproduction-related research findings through genome sequencing will ensure the expansion of genetic assessment into new areas such as genomic prediction of reproductive phenotypes, pharmacogenomics and molecular embryology, further enhancing our knowledge and therapeutic tools for treating infertility and improving women’s health [119].

### 2.6. Inflammatory Factors

Inflammation and infection are important factors in the progression of female reproductive disorders and infertility. Since many women are asymptomatic, especially upper genital tract infections often lack specific imaging features and laboratory tests, making difficult a definitive clinical diagnosis and increasing the risk of reproductive sequelae due to delayed treatment. As a result, the systematic evaluation of biomarkers can improve the sensitivity and specificity of the diagnosis of pelvic inflammatory diseases and sequelae assessment.

TroA and HtrA are proteins that are expressed during persistent chlamydia infection in vitro. Studies presented that TroA and HtrA serology positivity is more common in women with tubal factor infertility (TFI) than in women with other causes (TroA 45.5 vs. 19.1%, *p* = 0.004; HtrA 36.4 vs. 13.1%, *p* = 0.004) and can be developed as specific biomarkers for chlamydia trachomatis-associated TFI [120]. Zheng et al. [121] identified 21 genes that recognize asymptomatic sexually transmitted infection (STI)-induced endometritis and distinguish it from non-STI pelvic pain and other diseases. Inflammatory biomarkers such as serum amyloid A, alpha-1 acid glycoprotein, peroxiredoxin 4 (Prx4), trimethylamine-N-oxide (TMAO), interferons (IFN)-γ, C-reactive protein (CRP)/albumin ratio, IFN γ-induced protein 10 kDa (IP10/CXCL10), heme oxygenase-1 and procalcitonin (PCT) [122,123,124,125,126,127,128,129,130] are involved in pathophysiological processes such as sex hormones, insulin, cell proliferation, oxidative stress and lipid metabolism in PCOS, so they can be used as effective biomarkers to predict PCOS and to predict the risk of cardiovascular disease [131,132] and chronic low-grade inflammation in offspring [133]. A recent study [134] showed that the inflammatory reaction coordinated by nuclear factor (NF)-κB signaling is exacerbated by abnormalities in the estrogen receptor-β and progesterone receptor pathways, and that these pathways are also affected by local inflammation, creating a dysregulated inflammatory-hormonal circuit, providing new insights into the origin and pathogenesis of endometriosis. High binding of CCR1 to RANTES enhanced the recruitment of inflammatory cells at the site of endometriosis and correlated significantly and positively with the severity of dysmenorrhea, providing a potential biomarker for the pain and inflammatory response associated with deep-infiltrating endometriosis (DIE) [135]. Ovarian natural aging is significantly associated with immune cell infiltration and activation of inflammation-related signaling pathways, with inflammation levels reaching a maximum during early ovarian aging and then gradually decreasing. This provides a research basis for exploring the mechanisms of natural ovarian aging [136].

### 2.7. Single-Cell Omics and Multi-Omics

Conventional histology assumes that cells are homogeneous, but this completely contradicts the reality of cellular heterogeneity in the organization of biological systems. Sequencing results actually reflect the average of gene expression in a population of cells. Therefore, sequence and expression information at the single-cell level reveals genomic heterogeneity in biological samples at single-cell resolution, which can reveal complex heterogeneous mechanisms involved in disease onset and progression, thus further improving disease diagnosis, prognosis prediction and the monitoring of drug treatment effects [137]. The normal development of germ cells and embryos is the basis of life extension, and is a hot topic for research on infertility and other reproductive problems. The first application of single-cell sequencing technology was in germ cells (eggs and pre-implantation blastocyst) [138]. In 2013, Tang and Qiao’s group [139] used single-cell WGS to reveal the morphological and dynamic changes of chromosomes during oocyte development, and to map the precise genetic profile of individuals and reveal the chromosomal changes and characteristics of female germ cells during meiosis. In the same year, they also developed single-cell transcriptome sequencing technology to map the high-precision single-cell gene expression characteristics of human preimplantation embryos [140]. In 2018, they resolved the DNA methylationome atlas of human preimplantation embryonic development at the single-cell level [141]. In 2021, the profiling of DNA methylation levels and patterns at different stages of human fetal germ cells at single-cell resolution was carried out and revealed the epigenomic dynamics of germ cell development [142]. In addition, single-cell omics techniques have been widely applied in the field of reproduction, including oocyte aging [143], sex determination [144], the immune environment of endometriosis [145] and the signaling pathway transduction mechanism of thin endometrium [146].

Single-cell omics mainly reflect only one aspect of biological systems. Multi-omics allows a systematic understanding of information flow at different omics levels and can provide an overall view of the pathophysiology of a disease, helping people to understand living systems more fully, which is of great importance for research in the life sciences. However, current multi-omics studies mainly focus on the integration and analysis of different types and sources of data, which are divided into vertical and horizontal aspects according to the type of data integrated. Vertical integration is targeted at a disease to explore the complex interactions between multiple omics datasets, and to identify potential biological pathways that support early detection and prevention of complex diseases [147], whereas systematic investigations based on the entire regulatory network of genomics—transcriptomics—epigenomics—proteomics—metabolomics—microbiomics phenotypes have not yet been achieved, and currently just make medically relevant predictions for high-risk variants at two or three of these levels. Horizontal integration explores the association of various reproductive disorders or disorders of the reproductive system with other systems at the same omics level to understand shared pathogenesis and hence mechanisms underlying co-morbidity [148,149]. Depending on the methods used for data integration and analysis, multi-omics data integration can be based on statistical methods [150,151], ML algorithms [152,153,154] and deep learning neural networks [155,156,157], which have different focuses with their independent strengths and limitations. The current multi-omics data integration and analysis address the following research questions. Firstly, the unique characteristics [158] of samples, experimental methods and interventions during the data integration process, as well as the fragmented nature of the relevant biological knowledge [159], make data integration and cleaning extremely difficult. Secondly, although current integration analysis methods and algorithmic models have succeeded at a scale, they are still with limited integration capacity, as each dataset is just analyzed separately and then the final results are combined. Thirdly, technical biases are introduced during the data pooling that do not necessarily accurately reflect the true condition and interfere with the ability to study biological mechanisms. Therefore, based on the improvement of experimental techniques and the development of effective and efficient dataset integration methods or algorithmic models, setting up evaluation methods that are compatible with the differences between different omics data, mining knowledge and regulation implied in the data, and enhancing the interpretability of the results will be an important direction for the development of multi-omics.

## 3. Application and Development of AI in Reproductive Medicine

If identifying biomarkers is regarded as theory informatics, AI can contribute to practical informatics. Since the late 1990s, a growing number of researchers have sought to apply AI to reproductive medicine, mainly in the areas of sperm assessment, prediction of embryonic potential and ART pregnancy outcomes [160,161,162]. Traditionally, clinicians often judge sperm function according to the count, motility and morphology, and embryos are scored according to the number and size of blastomere, as well as the proportion of fragmentation. However, this way does not represent the developmental potential of gametes well and cannot exclude genetic abnormalities. PGT can effectively identify and exclude embryos carrying aneuploid or monogenic disease, however, its clinical implications for those without clear clinically acceptable indications has been controversial [163,164]. Meanwhile, its complex manipulations and invasive procedure may sometimes lead to wrong assessment [165]. Therefore, deep learning-based AI integrated in multi-subjects such as image segmentation, data mining, convolutional neural network (CNN), information on the molecular network and patient characteristics are becoming an important research direction for contemporary reproductive medicine (Figure 3). Hamme et al. [166] constructed a CNN model based on the unique morphological characteristics of each embryo for three replicates to train and validate a total of 4889 time-lapse embryo images from 400 patients on days 3 and 5. All were matched with 100% accuracy for patient identification in a random pool of eight patients’ embryo cohorts. Integration of this technique with imaging systems and laboratory protocols allows for improved sample tracking. Embryologists’ assessment of embryo quality can lead to significant differences in decision making due to subjective judgments. A 10-fold cross-validated deep neural network (DNN) was developed in Israel to provide implantation probability ratings for time-lapse video images. Logistic regression was applied to confounding variables to compare the accuracy and consistency of the model with embryologists’ predictions when assessing the probability of implantation of blastocysts. The AUC for DNN was higher than for embryologists as a whole (0.70 vs. 0.61) [167]. Cheredath et al. [168] incorporated day 5 embryo culture medium metabolomics data and associated embryological parameters into a custom artificial neural network (ANN) model to improve the predictability of embryo implantation potential. Several studies [161,162,169,170] have compared different ML algorithms such as logistic regression (LR), decision trees (DT), naive Bayes (NB), random forests (RF), support vector machines (SVM), neural networks (Nnet), back propagation neural networks (BNN), gradient boosting decision trees (GBDT), extreme gradient boosting (XGBoost) and super learners (SL) in pregnancy prediction and clinical decision making with good performance, especially SVM, RF and SL. When combined with molecular fingerprinting, it can also be used to predict the reproductive toxicity of chemicals [171]. Moreover, an AI-based image recognition and cloud computing sperm motility testing system achieved sperm analysis by smartphone at any time without the need for hospital visits, and more data can be collected to aid in clinical decision making and epidemiological studies [160]. The genetic background prediction is based on morphology and dynamics of gametes [172]. Predicting pregnancy outcomes in women with recurrent reproductive failure is based on maternal characteristics and the immune status of the body [173]. Assessment of endometrial receptivity based on weighted co-expression of existing web data mining [174] or experimental results [175]. In conclusion, in the field of reproductive medicine, AI has enormous potential for expansion. With the continuous progress and innovation of science and technology, it is believed that the combination of reproductive medicine and AI will transform the traditional medical model in terms of medical services and the doctor–patient relationship, and achieve “patient-centred” individual and precise healthcare.

### Limitations and Challenges of AI in Reproductive Medicine Applications

Although AI has made great progress in the field of reproductive biology, scientists are beginning to consider incorporating more clinical information, test variables and image data during neural network computing to make the calculations more accurate. However, real-world clinical applications have not yet become a reality. First, big data are a prerequisite for AI applications, but patient medical data are personal and face elevated technical and management risks in the data integration process, and centralized data storage increases the risk of leakage. Although laws and regulations for data storage [176] have been introduced in different countries as well as internationally, how data are used and protected is still an important topic. Second, the prerequisite for accurate computations in AI is standard data and metadata, so that the data are expressed in human-readable and machine-computable form, i.e., ontology [177]. Therefore, there is a need to develop effective and interoperable applications of ontologies for data and knowledge integration and to establish a standard reproductive medicine knowledge graph. Third, currently developed AI is mainly aimed at the assessment and prediction of ART [178], and there are still many vacancies in reproductive medicine such as prevention of male and female reproductive diseases and assessment of fertility risk, as well as how to precisely control the dose of drugs during superovulation and avoid ovarian hyporesponsiveness or ovarian hyperstimulation, which may be the future direction of AI.

## 4. The Way from Reproduction Medical Database to Knowledge Graph

Database construction is a key segment of advancing clinical medical research. A standardized, high-quality database is an essential foundation for subsequent statistical analysis, and is also the key to drive medical AI towards clinical application. Medical databases can be divided into three types in terms of function. The first type are medical information databases (MIDs); with informatization construction in hospitals, electronic medical records (EMRs) can be popularized in hospitals at all levels, providing the possibility of establishing MIDs [179]. This kind of database mainly consists of a patient’s clinical diagnosis and treatment process, laboratory tests and imaging examinations. It is convenient to extract medical information and systematically understand the process of disease development and medical cost calculation of patients. The second type are molecular databases; with the development of sequencing and mass spectrometry technology, a large amount of data has been generated. Therefore, in order to store and manage these molecular data, numerous public databases have been developed, which are also the main sources of basic medicine and bioinformatics research at present. The third type are disease databases; this kind of database is mainly a secondary database developed on the basis of the above two databases and other research results (e.g., literature, thesis). Researchers integrate key information from EMR, molecular data or existing research results according to a certain disease or drug, which helps deeper mining of phenotypes, molecular networks and interactions [180], and offers the possibility of building knowledge graphs. On this basis, introducing information technology such as AI and cloud computing is conducive to the context of health, the formation of medical guidelines and the integration of clinical decision making and health management (Figure 4).

### 4.1. Status of Reproductive Medicine Database

EMR systems of reproductive medicine are usually developed by different commercial companies, so there are problems such as different levels of system development and inconsistent information collection, making it difficult to integrate the wide variety of systems into a large multi-centre database and to promote data sharing. Secondly, regardless of the country, ART is under strict governmental supervision. As a result, national health authorities or professional societies collect and publish annual clinical and embryo laboratory data from each centre on the appropriate websites, but these data often require special permissions to access, and such statistical data are used more primarily for national and societal technical quality control and for setting industry standards, making it difficult to carry out more in-depth data mining. Apart from that, we can also retrieve reproduction-related information from other health or disease databases to study their relation to human reproduction. For example, Houtchens et al. [181] used real-world data from IQVIA (an administrative claims database) to explore the impact of multiple sclerosis on female live birth rates, infertility diagnosis and infertility treatment. Martin et al. captured reproductive toxicity data in the Toxicity Reference Database (ToxRefDB) [182] to analyze the reproductive toxicity of chemicals from multi-generations. Beyond that, reproduction-related databases are still scarce, with problems such as a single database structure, inadequate information and limited clinical guidance (Table 1). Thus, much exciting work remains to be carried out.

### 4.2. Constructing Clinical Reproductive Explainable Knowledge Graph Based on Ontology

Knowledge graph is a special kind of database for knowledge management and the provision of efficient knowledge services. It is a collection of knowledge based on a general database from which knowledge points are purposefully extracted and organized according to a certain knowledge system for orderly collation and analysis. Building a knowledge graph on the basis of a knowledgebase starts with semantic disambiguation of natural language, using ontologies as support for semantic differentiation and correct understanding of utterances; ontologies are then used to build a schema layer that provides a common understanding of relevant domain knowledge, highlighting and emphasizing concepts and the connections between concepts. Phenotypic information about diseases is often incomplete, inaccurate or even erroneous, which limits the accuracy and efficiency of phenotype-based analysis in diagnosing diseases. On the basis of large-scale, high-quality general knowledge graphs, diverse molecular, clinical and image information is collected, analyzed, evaluated, processed and stored according to certain themes, and various trivial and fragmented knowledge is interconnected in the form of graphs, turning disordered data into knowledge networks and quickly obtaining logical relationships between knowledge and knowledge, allowing for top-down and bottom-up deep phenotype mining, thereby classifying diseases with precise subtypes and effectively predicting progression and regression. At the same time, the construction of the explainable AI system avoids the inherent defects of algorithmic black boxes, improves the ability to analyze and solve problems, has the ability to diagnose and treat and can meet the personalized needs of diverse users and realize intelligent healthcare.

In conclusion, with the development of informatics technology, health and medical big data based on biology and omics are increasingly and systematically collected, with the features of massive data scale, high-speed data exchange and various data types. To identify characteristic biomarkers and their regulatory networks related to human reproductive disease through differential analysis of these big data. These biomics data and clinical data are fused across repositories, pools knowledge from large-scale electronic medical records, disease-related life-omics, third-party knowledge bases and other evidence sources, and builds a “gene-pathway-disease-symptom-treatment-drug” knowledge graph of precision medicine. Using explainable AI to improve the accuracy, reliability, causality, transparency, safety and fairness of knowledge, the system forms a disease-related knowledgebase for precision medicine.

## Figures and Tables

**Figure 1 ijms-24-00004-f001:**
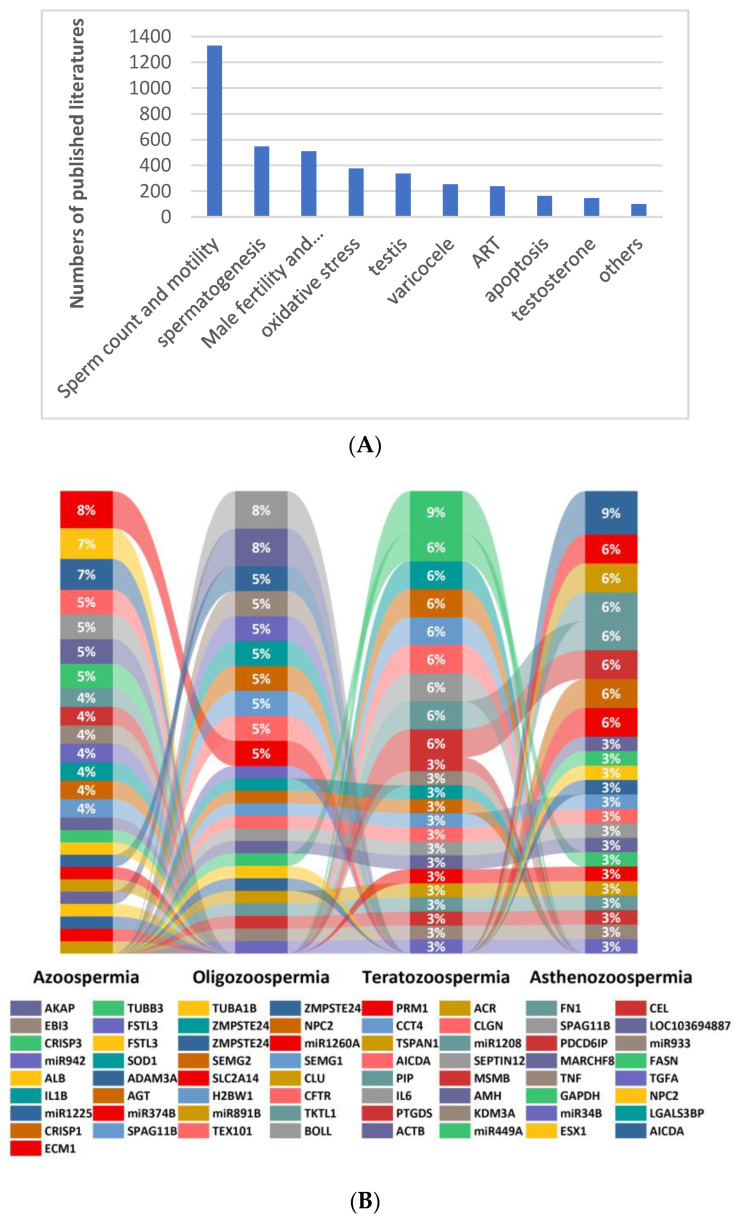
Major areas of research on biomarkers related to male infertility (**A**) and biomarkers commonly associated with male sperm abnormalities (**B**) (literature statistics from January 2012 to July 2022).

**Figure 2 ijms-24-00004-f002:**
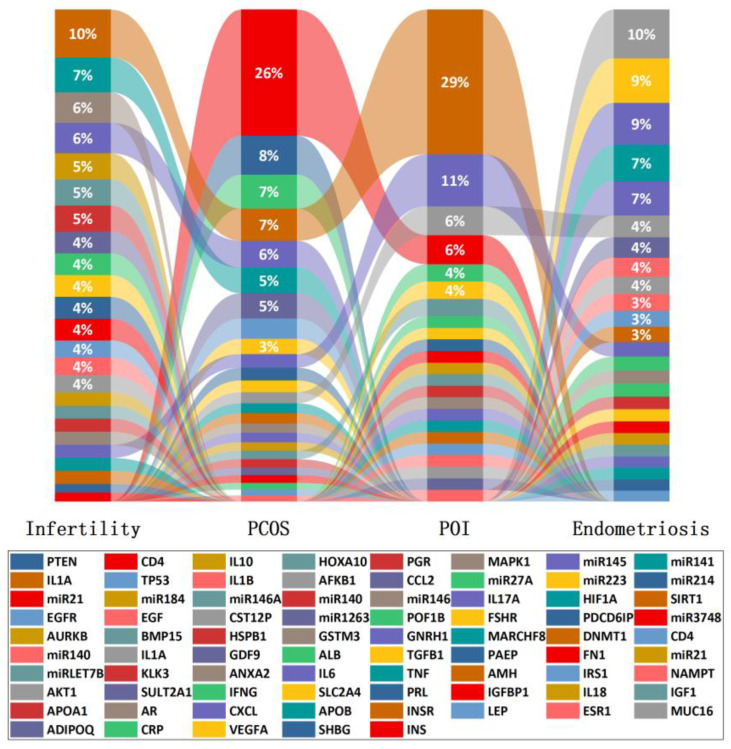
Biomarkers associated with common female reproductive disorders such as infertility, PCOS, POI and endometriosis and their proportion in the investigation of each condition.

**Figure 3 ijms-24-00004-f003:**
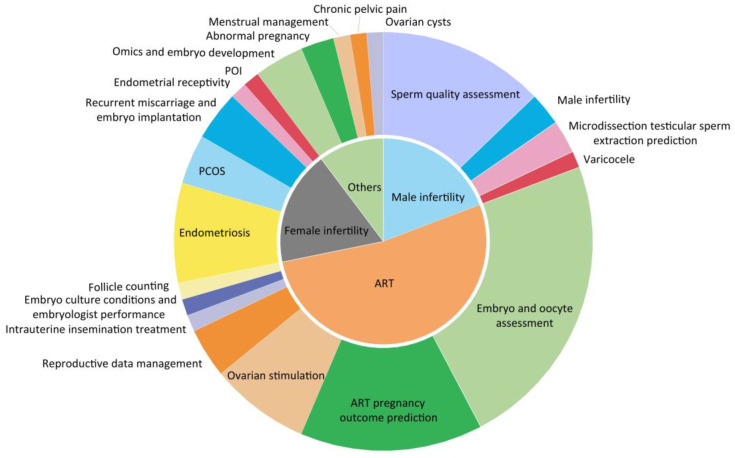
Application of AI in pregnancy prediction.

**Figure 4 ijms-24-00004-f004:**
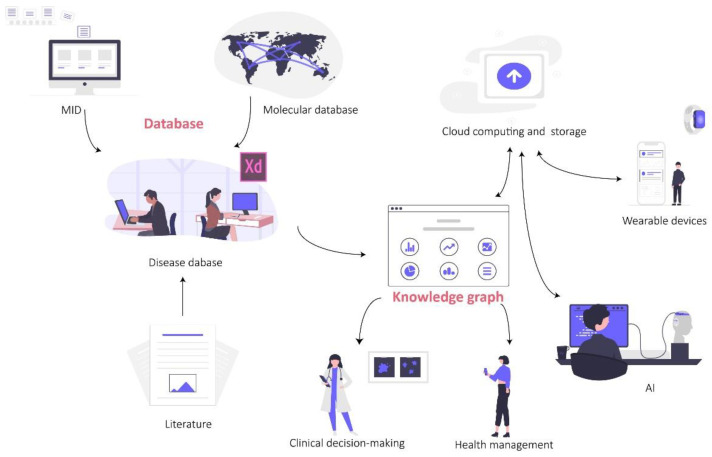
The application of database and knowledge graph.

**Table 1 ijms-24-00004-t001:** Examples of reproduction-related databases and knowledge graphs.

Database Name	Website	Development Agencies	Characteristic
Society for Assisted Reproductive Technology (SART)	https://www.sart.org	US ART Society	One of the largest reproductive medicine societies in the world, with over 90% of fertility centres in the US as members. Annual assisted reproduction statistics and industry standard setting in the US.
International Committee Monitoring Assissted Reproductive Technologies (ICMART)	https://www.icmartivf.org	International Conference Services	It takes a leading role in the development, collection and dissemination of worldwide data on ART through its World Report series.
Centers for Disease Control and Prevention (CDC)	https://www.cdc.org	US CDC	Annual assisted reproduction statistics and industry standard setting in the US.
European Society of Human Reproduction and Embryology (ESHRE)	https://www.eshre.eu/en	ESHRE	Annual assisted reproduction statistics and industry standard setting in Europe.
Human Fertilisation and Embryology Authority (HFEA)	https://www.bionews.org.uk	UK Department of Health	It is responsible for the regulation and inspection of all UK clinics offering in vitro fertilization, artificial insemination and human egg, sperm or embryo storage. It is also responsible for human embryo research.
Chinese Society of Reproductive Medicine (CSRM)	http://csrm1.meetingchina.org/msite/main/cn	Chinese Medical Association	Annual assisted reproduction statistics and industry standard setting in China.
Massachusetts Outcomes Study of Assisted Reproductive Technology (MOSART)	-	MGH Center for Child and Adolescent Health Research and Policy, MassGeneral Hospital for Children, US	It linked the SART Clinical Outcomes Reporting and the Massachusetts Pregnancy to Early Life Longitudinal (PELL) data systems, to provide a strong basis for further longitudinal ART outcomes studies. It also supports the continued development of potentially powerful linked clinical-public health databases [183].
The Catalog of Genes Associated with Different Forms of Lowered Semen Quality Caused by Impaired Spermatogenesis (HGAPat)	https://www.sysbio.ru/hgap/	Novosibirsk State University	A catalog of human genes associated with lowered semen quality (HGAPat) and analyzed their functional characteristics [184].
MeiosisOnline	https://mcg.ustc.edu.cn/bsc/meiosis/index.html	University of Science and Technology of China	A manually curated database for tracking and predicting genes associated with meiosis [185]
Male Fertility Gene Atlas (MFGA)	https://mfga.uni-muenster.de	Germany Centre of Reproductive Medicine and Andrology, University Hospital Münster	It enables a more targeted search and interpretation of OMICS data on male infertility and germ cells in the context of relevant publications [186].
SpermBase	http://www.spermbase.org	Department of Physiology and Cell Biology, University of Nevada School of Medicine, Reno, Nevada	A database for sperm-borne RNA contents [187]
GermlncRNA	http://germlncrna.cbiit.cuhk.edu.hk/	The Chinese University of Hong Kong	A unique catalogue of long non-coding RNAs and associated regulations in male germ cell development [76].
Dragon Exploration System for Toxicants and Fertility (DESTAF)	http://cbrc.kaust.edu.sa/destaf	King Abdullah University of Science and Technology (KAUST), Thuwal, Saudi Arabia	A database of text-mined associations for reproductive toxins potentially affecting human [188].
GermSAGE	http://germsage.nichd.nih.gov	Eunice Kennedy Shriver National Institute of Child Health and Human Development	A comprehensive SAGE database for transcript discovery on male germ cell development [77].
Reproductive and developmental toxicology (REPROTOX)	http://www.fda.gov/cder/Offices/OPS_IO/default.htm	US FDA	The database is suitable for QSAR modeling and human hazard identification of untested chemicals [189].
Male Infertility Knowledgebase (MIK)	http://mik.bicnirrh.res.in/	ICMR-National Institute for Research in Reproductive Health, India	A platform for review of genetic information on male infertility, identification pleiotropic genes, prediction of novel candidate genes for the different male infertility diseases and for portending future high-risk diseases associated with male infertility [64].
Endometriosis Knowledgebase	http://www.ek.bicnirrh.res.in	ICMR-National Institute for Research in Reproductive Health, India	The database includes genes, pathways, gene ontologies and and protein functions common to endometriosis [190].

CDC, Centers for Disease Control and Prevention; US, United States; FDA, Food and Drug Administration; ICMR, Indian Council of Medical Research; MGH, Mass General Hospital; QSAR, quantitative structure–activity relationship; UK, United Kingdom.

## Data Availability

Publicly available datasets were analyzed in this study. This data can be found here: [https://www.citexs.com/].

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
