# Peer review of "Translational Bioinformatics for Human Reproductive Biology Research: Examples, Opportunities and Challenges for a Future Reproductive Medicine"

_ijms, 2022, doi:10.3390/ijms24010004_

Round 1
Reviewer 1 Report
The authors describe and analyze the methods, applications and possibilities of bioinformatics in reproductive medicine and its research.
Form
The quite long essay is appropriate in length (text 13 printed pages) for its ambitious aim. The references appear too long on first sight, but appropriate after reading the paper. Figures and tables (3) are appropriate in number. Language is ok.
List of abbreviations: not necessary and also in part not correct, e. g. as ART is double and is not the abbreviation of assisted reproduction, EDC is not the abbreviation of electronic medical record. It is sufficient to be explained – correctly – in the text.
The authors should give their professional background.
Content
The authors use the term reproductive medicine uniformly for research and clinical practice, which is misleading.
So, the title is misleading, as the article and its literature is primarily dealing with research issues and not with every day medicine. About the first half of the references is dealing with prostate cancer and male infertility, all part of urology, whereas reproductive medicine is globally seen as one of three subspecialities of obstetrics and gynecology; the second half of the references primarily deals with diseases to be dealt with in reproductive medicine, but by far not in every case any of these: endometriosis, polycystic ovary and genetic problems.
So, the title better should be: Translational bioinformatics for human reproductive biology research: examples, opportunities and challenges for a future reproductive medicine.
None of the authors seem to be physician nor specialist in reproductive medicine. So, to write about a subject without practical competence is risky. The tenor of the article suggests that bioinformatics in terms of AI and its allies can and should be improved in reproductive medicine. But fact is – unfortunately – that in clinical practice it is not applied at all, up to now it is purely experimental.
So, page 9, 3., line 2: “ … AI has made remarkable progress in reproductive medicine…” is wrong. This is correct for research, but not for clinical practice. In the same paragraph: “… the most common AI-based morphological grading and prediction…” seems to suggest, that AI grading of sperms is common, but it is not. Further down: “… AI are driving massive transformations in reproductive medicine…” – this might be true for the future, but at the moment in clinical practice not yet! Next paragraph: “AI has greatly evolved in reproductive medicine in recent years…” – in research yes, but not in practice.
Table 1: The legend should read: “Examples of … databases…”, as from the few published ART registries in English the registries of the world (ICMART), Australia-New Zealand and Germany are missing.
The paper is interesting, informative and valuable to readers from research and clinical practice, with a sound analysis of all ingrediencies for AI application in reproductive medicine. Its views broaden the perspective of this medical subject, the trends are described correctly and future developments are seen in a very probable way. Clinics should get prepared for a new aera of computed aided medicine by AI with integration of clinical and laboratory data including all branches of omics for precision medicine also in reproductive medicine. The paper can serve as a guideline for those interested in this direction and therefore should be published.
Author Response
Dear Reviewer,
Thank you very much for your comments and professional advice. These opinions help to improve academic rigor of our article. Based on your suggestion and request, we have made correct modifications on the revised manuscript. We hope that our work can be improve again. Furthermore, we would like to show the details as follows:
Point 1: List of abbreviations: not necessary and also in part not correct, e. g. as ART is double and is not the abbreviation of assisted reproduction, EDC is not the abbreviation of electronic medical record. It is sufficient to be explained – correctly – in the text.
Response 1: Many thanks to you for helping us to point out the overlooked details, and we have checked and revised the abbreviations throughout the document. In the meantime, we have organized the abbreviations into a table, see the end of the article, to make it clearer. (Page 16~17)
Point 2: The authors should give their professional background.
Response 2: Because the journal International Journal of Molecular Sciences does not require introductions of authors and corresponding authors, there is no relevant content written. However, we are happy to provide professional introductions here.
Kun Liu, MD in Obstetrics and Gynaecology, PhD in Molecular Biology and Biomedicine, has been working in clinical related work in reproductive endocrinology and assisted reproduction technology for 18 years.
Yingbo Zhang, PhD in medical informatics, specialising in biomedical knowledge bases and artificial intelligence related to male sperm.
Cesar Martin, PhD. Associate Professor, Supervisor of Masters and PhD students. Academic Secretary of Molecular Biology and Biomedicine, University of the Basque Country, UPV/EHU. Permanent Researcher at Biofisika Institute. His main research is in clinical and basic research on molecular biology, combining artificial intelligence, bioinformatics, complex networks and systems biology for the derived creation of experimental optimization.
Bairong Shen, PhD, Professor. Supervisor of Masters and PhD students. Executive Director of the Institute of Disease Systems Genetics, West China Hospital, Sichuan University. He has been engaged in bioinformatics and translational medical informatics research and teaching for more than 20 years, and has accumulated many years of research experience in multi-omics informatics, especially in computer-aided biomarker discovery, multi-omics data fusion modeling and sharing, and 5G and smart health.
Xiaoling Ma, MD in Obstetrics and Gynaecology. Associate Professor, M.A. Supervisor. Director of Reproductive Medicine Centre, First Hospital of Lanzhou University. 20 years in reproductive medicine.
Point 3: The authors use the term reproductive medicine uniformly for research and clinical practice, which is misleading.
So, the title is misleading, as the article and its literature is primarily dealing with research issues and not with every day medicine. About the first half of the references is dealing with prostate cancer and male infertility, all part of urology, whereas reproductive medicine is globally seen as one of three subspecialities of obstetrics and gynecology; the second half of the references primarily deals with diseases to be dealt with in reproductive medicine, but by far not in every case any of these: endometriosis, polycystic ovary and genetic problems.
So, the title better should be: Translational bioinformatics for human reproductive biology research: examples, opportunities and challenges for a future reproductive medicine.
Response 3: Thank you again, for your careful review and for suggesting a very good topic. It is true that there was some struggle with the concept of reproductive medicine. For physicians who practice assisted reproductive medicine, the concept would be a narrow one, mainly around reproductive health issues related to infertility. But in a broader sense, reproductive medicine is about much more than that. So, thank you very much for your advice. (Page 1 line 1)
Point 4: None of the authors seem to be physician nor specialist in reproductive medicine. So, to write about a subject without practical competence is risky. The tenor of the article suggests that bioinformatics in terms of AI and its allies can and should be improved in reproductive medicine. But fact is – unfortunately – that in clinical practice it is not applied at all, up to now it is purely experimental.
Response 4: We introduced the members' specialties at Point 2. As mentioned, Kun Liu and Professor Ma worked for a long time in clinical work in reproductive medicine, and PhD Zhang worked on basic research in male reproduction and bioinformatics. Professors Martin and Shen both have very extensive experience in molecular biology and systems biology. The team expects to use the advanced methods of bioinformatics to promote in-depth research and development in reproductive medicine, and to achieve clinical translation is exactly our goal.
Point 5: So, page 9, 3., line 2: “ … AI has made remarkable progress in reproductive medicine…” is wrong. This is correct for research, but not for clinical practice. In the same paragraph: “… the most common AI-based morphological grading and prediction…” seems to suggest, that AI grading of sperms is common, but it is not. Further down: “… AI are driving massive transformations in reproductive medicine…” – this might be true for the future, but at the moment in clinical practice not yet! Next paragraph: “AI has greatly evolved in reproductive medicine in recent years…” – in research yes, but not in practice.
Response 5: Thank you for pointing out these areas that are prone to conceptual misinformation. We modify “… AI has made remarkable progress in reproductive medicine…” to “a growing number of researchers have sought to apply AI to reproductive medicine, mainly in the areas of sperm assessment, prediction of embryonic potential and ART pregnancy outcomes”. (Page 10, below point 3)
In this paragraph we have provided some additional examples of the machine learning applied to reproductive biology. As a result, “… the most common AI-based morphological grading and prediction…” has been removed. (Junction of page 10 and 11)
Further down: “… AI are driving massive transformations in reproductive medicine…” This section has been modified as follows, “In conclusion, in the field of reproductive medicine, AI has enormous potential for expansion. With the continuous progress and innovation of science and technology, it is believed that the combination of reproductive medicine and AI will transform the traditional medical model in terms of medical services and doctor-patient relationship, and achieve "patient-centred" individual and precise healthcare.” (End of first paragraph on page 11)
Next paragraph: “AI has greatly evolved in reproductive medicine in recent years…” was modified to “Although AI has made great progress in the field of reproductive biology” (Page 12, below Figure 3)
Point 6: Table 1: The legend should read: “Examples of … databases…”, as from the few published ART registries in English the registries of the world (ICMART), Australia-New Zealand and Germany are missing.
Response 6: Still grateful to you for your attention to many details and for the good advice you have given us. It is true that each country will have its own corresponding ART management system and we cannot list them all. We have therefore followed your suggestion. At the same time, ICMART makes sense as a large global data site incorporating data from several countries, so we have added it. (Page 14)
In conclusion, please allow us to express our gratitude to you once again. From your response, we have felt your scientific rigor and patience. Your pertinent suggestions will urge us to be more careful and conscientious in our future work.

Reviewer 2 Report
The major comments:
In the present study, authors the global review on the reproductive medicine and discussed on the impact and the limitation of collection and management of the comprehensive databases, including data-bases collecting risk exposures, disease markers for the development of translational bioinformatics for reproductive medicine. The manuscript, in general, is adequately written and provided basic information for the background. However, the information about how to performing artificial intelligence into reproductive medicine need more detail information and discuss.
1. The increasing availability of datasets combined with advances in machine learning and artificial intelligence, are facilitating a transformational advantage in reproductive medicine. Authors mentioned “deep learning-based AI integrated in multi-subjects such as image segmentation, data mining, convolutional neural network (CNN), information on the molecular network and patient characteristics are becoming an important research direction for contemporary reproductive medicine.” However, no detail application information for the ML algorithms and train models. Please include neural networks (CNN), artificial neural networks (ANNs), support vector machine (SVM), and Bayesian networks. In addition, please introduce the models for machine learning, deep learning, and ensemble learning, such as Logistic Regression, K nearest neighbor, Multi-Layer Perceptron, Decision Tree, and Deep learning model.
2. Please reorganize Figure 1 into three figures. The word size was too small and hard to reading. Figure 1b represent the etiology for infertility, not the most frequently reported biomarkers in female infertility.
3. No information for Figure 1C. Please give a complete description about how AI application in reproductive medicine in the Figure legend.
4. Several abbreviation words were shown without the full name, such as NOA and cfDNA. Please make an abbreviation table for all the abbreviations.
Author Response
Dear Reviewer,
Thank you very much for your comments and professional advice. These opinions help to improve academic rigor of our article. Based on your suggestion and request, we have made correct modifications on the revised manuscript. We hope that our work can be improve again. Furthermore, we would like to show the details as follows:
Point 1: The increasing availability of datasets combined with advances in machine learning and artificial intelligence, are facilitating a transformational advantage in reproductive medicine. Authors mentioned “deep learning-based AI integrated in multi-subjects such as image segmentation, data mining, convolutional neural network (CNN), information on the molecular network and patient characteristics are becoming an important research direction for contemporary reproductive medicine.” However, no detail application information for the ML algorithms and train models. Please include neural networks (CNN), artificial neural networks (ANNs), support vector machine (SVM), and Bayesian networks. In addition, please introduce the models for machine learning, deep learning, and ensemble learning, such as Logistic Regression, K nearest neighbor, Multi-Layer Perceptron, Decision Tree, and Deep learning model.
Response 1: This part of AI has been described in general terms, with few examples, and we have supplemented it accordingly. As follows: Hamme et al. constructed a CNN model based on the unique morphological characteristics of each embryo for three replicates to train and validate a total of 4889 time-lapse embryo images from 400 patients on days 3 and 5. All were matched with 100% accuracy for patient identification in a random pool of eight patients' embryo cohorts. Integration of this technique with imaging systems and laboratory protocols allows for improved sample tracking. Embryologists' assessment of embryo quality can lead to significant differences in decision-making due to subjective judgments. A 10-fold cross-validated deep neural network (DNN) was developed in Israel to provide implantation probability ratings for time-lapse video images. Logistic regression was applied to confounding variables to compare the accuracy and consistency of the model with embryologists' predictions when assessing the probability of implantation of blastocysts. The AUC for DNN was higher than for embryologists as a whole (0.70 vs. 0.61). Cheredath et al. incorporated day 5 embryo culture medium metabolomics data and associated embryological parameters into a custom artificial neural network (ANN) model to improve the predictability of embryo implantation potential. Several studies have compared different ML algorithms such as logistic regression (LR), decision trees (DT), naive Bayes (NB), random forests (RF), support vector machines (SVM), neural networks (Nnet), back propagation neural networks (BNN), gradient boosting decision trees (GBDT), extreme gradient boosting (XGBoost) and super learners (SL) in pregnancy prediction and clinical decision making with good performance, especially SVM, RF and SL. When combined with molecular fingerprinting, it can also be used to predict the reproductive toxicity of chemicals. (Junction of pages 10 and 11)
With regard to the introduction of the model, we consider the ultimate aim of this paper to be a discussion of clinical translation, rather than a specialised introduction, so we have not made a specialised description here. However, thank you very much for the very specialised advice you have given us.
Point 2: Please reorganize Figure 1 into three figures. The word size was too small and hard to reading. Figure 1b represent the etiology for infertility, not the most frequently reported biomarkers in female infertility.
Response 2: The three figures do look crowded in one place and the font does not show well. We have shown the three diagrams separately and placed them in the corresponding paragraphs to give a more coherent reading. Furthermore, we have replaced the original figure 1b (now figure 2) from a tree plot to a strip plot. On the one hand, this is because the strip plot font appears more clearly. On the other hand, it is possible to understand the biomarkers that play a common role between the different female reproductive diseases. (Page 5, 7 and 12)
Point 3: No information for Figure 1C. Please give a complete description about how AI application in reproductive medicine in the Figure legend.
Response 3: We apologize that the previous figure descriptions did not clearly break up a,b,c, which made them difficult to read. We have made the graphs apart and described them separately in detail.
Point 4: Several abbreviation words were shown without the full name, such as NOA and cfDNA. Please make an abbreviation table for all the abbreviations.
Response 4: Many thanks to you for helping us to point out these overlooked details and we have revised and made the abbreviations table. (Page 16, 17)
Finally, allow us to express once again our gratitude to you. Firstly, it is the rigorous attention to detail that drives us to improve ourselves. Secondly the more detailed explanation of artificial intelligence has also given us many new insights.

Round 2
Reviewer 2 Report
Dear Authors,
1. Please notify the Y-axis of the Figure 1 to be the numbers of published literatures or biomakers.
2. According the new revised Figure 2 for female infertility, please also prepare one figure for the biomarkers of male inferilty that will provide more understanding and insight for male infertility.
Author Response
Dear Reviewer,
Thank you very much for giving us further constructive advice, which is very important in improving the quality and integrity of the manuscript. We are pleased to follow your thoughtful suggestions regarding the figures in the manuscript with the corresponding changes and updates.
Point 1: Please notify the Y-axis of the Figure 1 to be the numbers of published literatures or biomakers.
Point 2: According the new revised Figure 2 for female infertility, please also prepare one figure for the biomarkers of male inferilty that will provide more understanding and insight for male infertility.
Response 1 and 2: Considering that both graphs are related to male infertility, we have included them both in Figure 1. The original Figure 1 is now labelled A (adding the notation " Numbers of published literatures" to the Y-axis) and the newly added figure is labelled B (biomarkers commonly associated with male sperm abnormalities).
Thank you again, Professor. Your tirelessness and dedication to excellence are of great importance in ensuring the quality and rigor of our research, and are a driving force for us to continue to improve ourselves.
